# Evaluating the Chemical Composition and Antitumor Activity of *Origanum vulgare* ssp. *hirtum* Essential Oil in a Preclinical Colon Cancer Model

**DOI:** 10.3390/ijms26104737

**Published:** 2025-05-15

**Authors:** Georgios Aindelis, Katerina Spyridopoulou, Sotiris Kyriakou, Angeliki Tiptiri-Kourpeti, Mihalis I. Panayiotidis, Aglaia Pappa, Katerina Chlichlia

**Affiliations:** 1Department of Molecular Biology and Genetics, School of Health Sciences, Democritus University of Thrace, 68100 Alexandroupolis, Greece; g.aindelis@gmail.com (G.A.); aikspiridopoulou@gmail.com (K.S.); mbg_tiptiri@yahoo.gr (A.T.-K.); apappa@mbg.duth.gr (A.P.); 2Department of Cancer Genetics, Therapeutics & Ultrastructural Pathology, The Cyprus Institute of Neurology & Genetics, Nicosia 2371, Cyprus; sotirisk@cing.ac.cy; 3Department of Comparative Biomedical Sciences, School of Veterinary Medicine, Mississippi State University, Starkville, MS 39762, USA; mp2358@msstate.edu

**Keywords:** *Origanum vulgare*, *hirtum*, essential oil, colon cancer, oral administration, antitumor immunity, apoptosis

## Abstract

*Origanum vulgare* ssp. *hirtum* is an aromatic plant native to various Mediterranean regions and has been traditionally used in folk medicine. This study investigates the chemical composition and the potential antitumor activity of its essential oil in a preclinical model of CT26 colorectal cancer in BALB/c mice. Mice received prophylactic oral administration of the essential oil, and tumor progression, immune modulation, and apoptosis were evaluated. Even treatment with low doses (350 parts per million, ppm in 100 μL final volume) of the essential oil significantly suppressed tumor growth by approximately 44%. This effect correlated with the enhanced expression of antitumorigenic cytokines, including a 2.7-fold increase in type I interferons (IFN), IFN-γ (from 46.5 to 111.9 pg/μL per mg of protein) and tumor necrosis factor alpha (TNF-α) (from 34.5 to 103 pg/μL per mg of protein). Furthermore, the production of granzyme B, a key mediator of cytotoxic immune cell function, was notably increased from 96.1 to 319.6 pg/μL per mg of protein. An elevated activation of caspase 3, a central effector caspase of all apoptotic cascades, was also observed in tumors from oregano-treated mice. These findings suggest that *O. vulgare* ssp. *hirtum* essential oil exhibits promising antitumor properties through immune modulation and immunity-mediated apoptosis induction, supporting its potential development as a bioactive compound for cancer prevention or therapy.

## 1. Introduction

*Origanum vulgare* ssp. *hirtum* is a member of the plant family *Lamiaceae*, comprising aromatic plants with medicinal properties. This particular variety, predominantly found in Greece, Cyprus, Turkey, and Italy [1,2], is known for its exceptional quality and high essential oil concentration and has been traditionally used as an treatment for respiratory disorders, stomach ache, painful menstruation, rheumatoid arthritis and has also been shown to have antimicrobial activity [3,4,5]. An intriguing recent observation has been the antiproliferative and pro-apoptotic effect of various *Origanum* species against cancer cells in vitro [6,7,8,9,10,11]. In addition, our group and others have reported the growth inhibitory and antimetastatic activity of extracts from other *Origanum* ssp. in preclinical models of colon and lung cancer. This particular variety of the plant, however, has not been explored in this manner [6,8,11].

Traditionally, natural, plant-derived products have been used against a number of ailments [12]. More recently, the worldwide prevalence of cancer combined with the serious side effects of treatment and the resistance of patients to chemotherapeutic agents has made the identification of novel anticancer compounds necessary [13]. Paclitaxel (Taxol) was the first chemotherapeutic compound derived from plants and up to this day remains a commonly used drug [14,15]. These efforts continue and several metabolites originating from plants, such as alkaloids, terpenoids, phenols, and flavonoids have been reported to display antitumor properties, including reducing the proliferation of cancer cells and inducing apoptosis [16,17], suppressing the migration and invasiveness of tumor cells [18], and inhibiting angiogenesis [16].

The essential oil extracted from *Origanum vulgare* ssp. *hirtum* has been shown to be comprised mostly of monoterpenes thymol. *p*-cymene, *γ*-terpinene, and carvacrol [5]. All of these compounds have been shown to induce antiproliferative and antitumor effects [19,20,21]. In general, monoterpenes have been widely investigated and studies have highlighted their cytotoxic and anticancer properties; however, limitations such as their lipophilicity and need for high concentrations in order for them to be functional have also been recognized [22,23]. Furthermore, extracts of *Origanum* ssp. are rich in other phenolic compounds [24,25]. These molecules like benzoic and cinnamic acids have been identified as potent anticancer mediators, with phenolics also shown to enhance antitumor immunity and interfere with tumor-induced angiogenesis [26,27].

The purpose of this study was to investigate the potential antitumor effect of various preparations of the essential oil extracted from the plant *Origanum vulgare* ssp. *hirtum*, chemically characterized for its composition, in a preclinical colon cancer cell model. Specifically, we examined the growth inhibitory effects of orally administered essential oil and its potential to eliminate tumor cells through the induction of apoptosis.

## 2. Results

### 2.1. Characterization of the Chemical Composition of Origanum vulgare ssp. hirtum Essential Oil

The chemical composition of the isolated oregano oil was analysed using UPLC-tandem mass spectrometry (MS/MS). Chromatographic conditions, such as mobile phase composition, flow rate, and elution, were optimised accordingly in order to acquire greater sensitivity for all the analytes. The optimisation of the mobile phase was based on multiple elution trials according to which separation efficacy of several solvent combination systems (including methanol/water and acetonitrile/water in various percentages) was examined. However, it was discovered that none of these combinations enhanced the peak’s symmetry and form. The acidification of water with 0.1% formic acid yielded peaks with considerably improved symmetry and shape [28]. Furthermore, the compounds’ ionisation was aided by the addition of formic acid. Further improvement of all peaks was achieved by raising the column temperature to 30 °C [29]. For the ionization of the various polyphenols that were presented in the oils, electrospray ionisation with either a negative or positive (ESI^±^) mode was employed. The International Counsil for Harmonization (ICH) requirements were followed for the validation of the analytical method [30]. In particular, linearity, precision, accuracy, limit of quantification (*LoQ*), and limit of detection (*LoD*) were all established. A linear regression equation was used to illustrate the standards-produced calibration curves.

A linear regression equation of peak areas vs. different concentrations ranging from 0.65 to 510 parts per billion (ppb) was used to illustrate the standards-generated calibration curves (Appendix A). In the range of 0.65 to 505.6 ppb, all polyphenols showed good linearity, whereas all of the standards under analysis had correlation coefficients (R^2^) more than 0.999 (Appendix A). Lastly, by calculating the recovery percentage, we assessed the UPLC-QqQ-ESI-MS/MS method’s repeatability. To achieve this, combinations of standard solutions of different polyphenols were added to the methanol solution of HEE. The findings of at least six repetitions were obtained from the preparation of spike samples in duplicate.

The % recovery was calculated according to Equation (1):(1)$$\%\;recovery=\frac{\left[\left(A-A_{0}\right)\right]}{A_{a}}\times 100$$
where A is the final amount detected, A_0_ is the initial amount, and A_a_ is the added amount.

Overall, our results revealed that *Origanum vulgare* ssp. *hirtum* isolated oil contained significantly high amounts of both flavonoids (17,899.24 ± 235.58 μg of catechin eq/g of dry extract) and phenolic compounds (25,600.65 ± 148.25 μg of gallic acid equivalents/g of dry extract) (Table 1), with eugenol (1060.20 ± 23.21 ng/g of dry extract) and kaempferol (3429.59 ± 89.5 ng/g of dry extract) being the most predominant polyphenol and flavonoid, respectively (Table 2).

Furthermore, we detected levels of condensed tannins (568.10 ± 15.88 μg of catechin equivalents/g of dry extract) and total monoterpenoid (12.42 ± 3.21 μg of linalool eq/g of dry extract). The content of polyphenolics (including benzoic, gallic, cinnamic, coumarin, phenolic acid, and furanocoumarins) and flavonoids (including flavanones, flavanols, and procyanidin) in *Origanum vulgare* ssp. *hirtum* were also evaluated using UPLC-ESI-MS/MS. A comparison of MRM transitions of standard compounds was utilized for the identification of the peaks (Table 2). According to our results, it was evident that the isolated oil is enriched in m-hydroxybenzoic acid (63.16 ± 2.47 ng/g of dry extract), ethyl gallate (127.04 ± 6.14 ng/g of dry extract), chlorogenic acid (262.55 ± 10.55 ng/g of dry extract), and 4′-methoxyflavanone (49.79 ± 2.21 ng/g of dry extract).

### 2.2. Tumor Growth Inhibition Following Consumption of Origanum vulgare ssp. hirtum Essential Oil

In order to evaluate the tumor inhibitory activity of the essential oil, we employed an ectopic syngeneic colorectal cancer model in BALB/c mice (Figure 1A). Mice were administered prophylactic doses of the essential oil daily via oral gavage prior to CT26 cell inoculation and during the initial days of tumor development, while control mice received only corn oil.

Animals were monitored throughout the experiment and no signs of discomfort or weight loss were observed (Appendix A). Furthermore, there were no differences in the spleen and liver indices of oregano-treated mice (Appendix A). A significant reduction in tumor volume was observed in animals that had received the essential oil, with tumors measuring up to approximately 80% smaller compared with those in control mice (Figure 2A,B).

In order to confirm that consumption of the essential oil was not toxic, we orally administered both the dose used in our tumor model (0.348 mg/kg of body weight) and a higher (7.5-fold) dose for 3 or 10 days in mice not harboring tumors and then measured serum levels of aspartame transaminase (AST/SGOT), alanine transaminase (ALT/SGPT), and alkaline phosphatase (ALP). No increase in any of the enzymes was observed in mice receiving the same amount of essential oil as animals in the tumor model. SGPT levels were found to have notably increased from 78.8 U/L to 132.2 U/L, but only when mice were administered with the significantly higher dose (Appendix A).

### 2.3. Expression Analysis of Cytokines and Tumor-Associated Molecules Following Administration of the Essential Oil

To investigate the mechanism by which the essential oil contributed to tumor growth inhibition, we first measured the circulating levels of IL-12, TNF-α, and IFN-γ in sera. However, no significant differences were observed between the treated and control groups (Appendix A). We then focused on the tumor tissue and evaluated the infiltration of immune cells. Surprisingly, once again, we could not detect any differences in the mice that had received the essential oil (Appendix A).

In contrast, cytokine analysis within the tumor microenvironment revealed a significant increase in TNF-α levels (Figure 3A), which was further supported by elevated TNF-α gene expression (Figure 3B). IL-12 levels remained unchanged, while IFN-γ was increased, but this upward trend did not reach statistical significance. Notably, granzyme B production in the tumor microenvironment was nearly doubled in the essential oil-treated group (Figure 3A). Given that granzyme B is a key effector molecule in the antitumor response of cytotoxic T lymphocytes and natural killer (NK) cells, this finding highlights a potential mechanism underlying the observed tumor suppression. Furthermore, we examined the expression of various cytokine and regulatory genes and observed upregulation of chemokine (C-X-C motif) ligand 10 (CXCL10) and IFN-α2 expression in the tumors of essential oil-treated mice (Figure 3B).

Next, we evaluated the expression of cyclooxygenase 2 (COX-2) and Ki67 in tumor cells (Figure 3C,D). COX-2 is an immunomodulatory factor associated with cancer cell adaptation for the suppression of the immune response, while Ki67 is a characteristic marker of cellular proliferation. A reduction in the proportion of tumor cells positive for COX-2 in that treated mice was observed, although the overall levels of COX-2 in the tumor were not significantly reduced (Figure 3E,F). Finally, we assessed the elimination of tumor cells by examining the activation of caspase 3, the main effector caspase in apoptosis. The percentage of tumor cells positive for caspase 3 activation increased significantly, from approximately 20% in control mice to more than 50% in oregano-administered mice (Figure 3C,D), while caspase 3 cleavage was more prominent (Figure 3E,F). To confirm the efficacy of cleaved, activated caspase 3, we also examined the cleavage of PARP1, a characteristic target of caspase 3 (Figure 3E,F).

### 2.4. Sustained Growth Inhibitory Effect of Oregano Essential Oil Following Long-Term Prophylactic Administration of Low Essential Oil Concentration

Our next approach focused on reducing the amount of essential oil administered per mouse. We also took into consideration the poor solubility of the essential oil in water-based solutions. As a result, we decided to evaluate its effects in the form of an emulsion dispersed in tomato juice and extended the period of prophylactic administration to compensate for the reduced oil intake (Figure 1B). Tumor growth was again inhibited in mice treated with the supplemented tomato juice for 41 days, with a reduction of approximately 44% in mean tumor volume (Figure 4). This effect was less pronounced than that observed with the concentrated essential oil, but it remained potent and statistically significant.

### 2.5. Impact of Low-Dose Oregano Emulsion on the Expression of Immunomodulatory Molecules in the Tumor: Comparable to High Concentrations of Essential Oil

To verify whether the mechanism underlying the activity of the diluted emulsion was the same as the concentrated essential oil, we examined the expression of the same molecular features. Our analysis focused on the tumor microenvironment, where the most pronounced differences had previously been observed. We detected a similar pattern of elevated expression and production of TNF-α in the tumors of treated mice (Figure 5A,B), as well as upregulated *CXCL10* and *IFN-α2* gene expression (Figure 5B). Additionally, the most important observation—the increased production of granzyme B—was persistent and even more prominent in this setting, with granzyme B concentration in emulsion-treated mice being almost three times higher than in the control group (Figure 5A).

Notably, we also detected a statistically significant increase in IFN-γ production in the tumors of these same mice (Figure 4A). Furthermore, the expression of COX-2 was significantly reduced in oregano-treated animals (Figure 5C,D). Finally, as before, to confirm that these cytotoxic elements were driving cancer cell killing, we evaluated the activity of caspase 3. We detected increased levels of the cleaved, active fragment of caspase 3 in the tumors of the treated group, along with activation through cleavage of its downstream target PARP1, indicative of apoptotic cell death and cancer cell elimination (Figure 5C,D).

## 3. Discussion

The use of medicinal plants has been a traditional practice throughout human history [12]. The great diversity of the secondary metabolites of various chemical groups in plant extracts contributes to many health beneficial effects, and has been translated in modern anticancer treatment with the design of medical products, such as the widely used chemotherapeutic drug Paclitaxel (Taxol) [14,15,16,17,18]. *Origanum vulgare* ssp. *hirtum* is an aromatic plant found in the Mediterranean basin, known for its high concentration of secondary metabolites and many biological activities, including antimicrobial, analgesic [3,4,5], and more recently identified anticancer properties, at least in in vitro systems [6,7,8,9,10]. The aim of this study was to chemically characterize the composition of *O. vulgare* ssp. *hirtum* essential oil and to evaluate the antitumor properties in a preclinical mouse model of colorectal cancer.

The composition of the essential oil was investigated using UPLC-MS/MS analysis and we identified kaemphenol, eugenol, and chlorogenic acid as its main constituents. Eugenol is a bioactive phenolic component in various aromatic plants [31]. Intriguingly, eugenol has been reported to reduce the proliferative capacity of different types of cancer cells, inducing cell cycle arrest and the apoptosis of tumor cells [32,33]. In addition, it has been shown to downregulate the production of cyclooxygenase 2 as well as inhibit angiogenesis and suppress the migration of cancer cells [33,34]. Chlorogenic acid is a phenolic acid synthesized by many aromatic plants, including coffee and tea [35]. Similarly to eugenol, chlorogenic acid has been identified as an antiproliferative and pro-apoptotic agent against cancer cells [36,37]. Moreover, it has been reported to inhibit matrix metalloproteinases expression, with these proteolytic enzymes being critical in tumor-mediated angiogenesis and immune evasion [37]. Perhaps more importantly, it has been shown to modulate antitumor immunity, upregulating genes such as the nuclear factor of activated T cells 2 (NFATC2) and NFATC3 that regulate TCR signalling and T cell activation [36,38], as well as favouring the polarization of M1 macrophages instead of M2 macrophages in a preclinical glioblastoma model [39].

Next, we decided to investigate the potential antitumor activity of the essential oil. To this end, we utilized an ectopic model of CT26 cancer cells injected into the back of BALB/c mice. Prophylactic oral administration of *Origanum* essential oil significantly inhibited the growth of the tumors. Intriguingly, this effect was persistent, although less prominent, with either short-term prophylactic administration of a high concentration of essential oil directly, or after long-term pre-emptive consumption of low doses of an emulsion, suitable for incorporation in water-based food ingredients. Similar observations have previously been made for extracts from other *Origanum* ssp., not only in colorectal cancer [6], but also lung cancer, where it has also been shown to suppress metastasis [8,11]. In addition, we did not observe any indication of adverse effects on the treated animals by either short- or long-term administration, as neither their body weight nor their spleen and liver indices were different from those of control mice. Additionally, serum levels of SGOT, SGPT, and ALP were not impacted by the consumption of the examined doses of the essential oil. Generally, the essential oil extracted from *Origanum vulgare* ssp. *hirtum* is considered safe for consumption for humans and animal species, inducing some variation in certain metabolic measurements, but no noticeable side effects or harmful outcomes [40,41,42].

In order to understand the underlying mechanisms mediating the growth inhibitory effect of the essential oil, we began exploring components of immune responses. We did not detect any differences in cytokines associated with improved antitumor immunity systemically. Even more surprisingly we did not observe any accumulation of immune cells capable of eliminating cancer cells, such as CD8 T cells or NK cells in tumors. However, tumor cell killing increased in response to essential oil administration. Both the percentage of cancer cells positive to caspase 3 activation and the overall levels of caspase 3 cleavage increased, as were the cleavage and deactivation of PARP1. These observations are indicative of elevated apoptotic activity against tumor cells [43,44]. Pro-apoptotic signaling and initiation is one of the major mechanisms employed by cytotoxic immune populations when targeting tumor cells [45]. More importantly, we discovered a notable increase in granzyme B production in mice that had received the essential oil. Granzyme B is a serine protease generated by NK cells and CD8 T cells and employed to induce apoptosis in their target cells. Caspase 3 is a substrate of granzyme B, acting as a mechanism for the direct activation of the execution phase of apoptosis, while granzyme B can also induce mitochondrial membrane integrity loss, leading to the activation of initiation pathways as well [46]. The release and subsequent fusion of perforin/granzyme B-containing granules is utilized by cytotoxic T cells in order to destroy tumor cells [45,47,48].

Furthermore, a modulation of immune function in the tumors of oregano-treated mice was observed, favoring the elimination of cancer cells. Noteworthy, the production of TNF-α and IFN-γ was enhanced in response to essential oil administration. These cytokines are associated with both the differentiation, migration, and activity of tumor suppressive T cell populations as well as other antitumor immune modifications [49,50]. Combined production of TNF-α and IFN-γ from CD4 T cells has been shown to alter the tumor microenvironment sensitizing tumors to various types of chemotherapy, such as treatment with cyclophosphamide and 5-fluorouracil [51]. In addition, IFN-γ promotes antigen presentation, as well as the upregulation of pro-apoptotic receptors and signaling molecules [52].

An increased expression of IFN-α2 was also detected in the tumors of essential oil-treated mice. IFN-α2 is a type I interferon, originally identified as a potent antiviral agent [53]. However, antitumor activities were later reported as well. These effects include the direct inhibition of tumor cell proliferation, modulation of tumor cell metabolism, enhancement of antigen presentation, and increased tumor cell recognition by CD8 T cells [53,54]. Moreover, type I IFNs act on immune cell populations, such as dendritic cells, stimulating their migration into lymph nodes and their accumulation in the tumor. They can also interfere with the suppressive activity of regulatory T cells and even promote their differentiation into Th17 T cells [53,55]. Another upregulated gene in the tumors of treated mice was that of *CXCL10*. CXCL10 is a small chemokine, induced by IFN-γ, primarily known for its role as an attractant for various immune cells [56,57]. However, the role of CXCL10 is not restricted in the mobilization of cells. It has been shown to amplify the activation of antigen-presenting cells, like dendritic cells or macrophages, promote the maturation of CD8 T cells, as well as increase the production of perforin and granzyme B, thereby improving the efficacy of immunotherapeutic treatments [57,58,59]. It is worth noting that, in addition to being induced by IFN-γ, CXCL10 has also been reported to be promoted by IFN-α. It may synergize with IFN-α, possibly through M1 polarization of tumor macrophages, to augment the cytotoxic activity and persistence of CD8 T cells [60].

In conclusion, our results indicated an immunomodulatory antitumor effect of the essential oil derived from *Origanum vulgare* ssp. *hirtum*. In this study, we demonstrated that even the long-term administration of low doses of the essential oil significantly impaired tumor growth in a preclinical colon cancer model. Notably, consumption of the essential oil altered the profile of cytokines and chemokines, such as IFN-γ, IFN-α2, and CXCL10, in the tumor, and enhanced the cytotoxic activity of immune cells. This culminated in the upregulated production of granzyme B, induction of apoptotic mechanisms, perhaps mediated by immune populations, through the activity of granzyme B and increased killing of cancer cells. Taken together, these observations support the hypothesis that this extract of *Origanum vulgare* ssp. *hirtum* enhances antitumor immunity against colorectal cancer and highlights its potential use, or that of its constituents, as an adjuvant nutraceutical agent with health beneficial properties. However, further studies are warranted in order to elucidate the exact mechanisms and molecular pathways involved in the activity of the essential oil.

## 4. Materials and Methods

### 4.1. Solvents

LC-MS grade solvents, such as chloroform (319988), methanol (34966), acetonitrile (34967), and formic acid (14265), were purchased from Honeywell (Charlotte, NC, USA). Chemicals: potassium dihydrogen phosphate (1.37039)_,_ potassium chloride (P3911), aluminium trichloride (563919), sodium acetate (S8750), and magnesium sulphate (63136) were purchased from Sigma Aldrich (St. Louis, MI, USA). Analytical standards: m-hydroxy benzoic acid (6098A), protocatechuic acid (6050), gallic acid (4993S), ellagic acid (6075), p-coumaric acid (6030), coumarin (0507S), chlorogenic acid (4991S), 4-methoxyflavanone (1186), naringin (1129S), isorhamnetin (1120S), quercetin-3-*O*-rhamnoside (1236S), myricetin-3-*O*-galactoside (1355S), myricetin-3-*O*-rhamnoside (1029S), kaempferol (1124S), and procyanidin-B2 (0984) were of >99% purity and purchased from Extrasynthese (Lyon, France). Assay kits: polyphenol quantification (Folin–Ciocalteu) (KB03006) was purchased from Bioquochem (Asturias, Spain).

### 4.2. Extraction of Essential Oil and Preparation of Oregano Mixtures

The essential oil was extracted from dried leaves by hydrodistillation at the facilities of Vioryl Chemical and Agricultural Industry, Research S.A. (Athens, Greece) and chemical composition was analyzed with GC/MS, as described in [4]. An emulsion containing the essential oil was prepared, as well as emulsions with medium-chain triglycerides, but no essential oil, to be used as controls [4]. The tomato juice–emulsion mixture was prepared in-house, adding both the emulsion with and without essential oil in a commercially available tomato juice, taking care not to exceed the maximum allowed concentration of the carrier. The final concentration was 350 ppm.

### 4.3. Preparation of Standards and Samples

Stock solutions of all the analytes were prepared in either an acetonitrile/water mixture (1:1) or methanol/acetonitrile mixture (1:1) at a concentration of 1000 ppm. Working standard solutions were made by diluting the individual standard stock solutions with ice-cold methanol. The isolated oil was diluted with ice-cold methanol at a final concentration of 100 ppb. Each solution was kept in darkness and protected from light in amber vials to minimize the auto-oxidation of polyphenols. In addition to this, stock, standard, and sample solutions were stored at −20 °C before use. All prepared solutions were passed through a 0.22 μm (mixed cellulose esters—MCE) membrane filtered prior to UPLC-QqQ-ESI-MS/MS analysis.

### 4.4. Liquid Chromatography Tandem Mass Spectrometry (LC-MS/MS) Conditions

For the chromatographic separation of the isolated extracts, a Waters Acquity UPLC system (Waters Corp., Milford, MA, USA) was employed. The separation was performed on an ACQUITY UPLC BEH C18 (100 × 2.1 mm, particle size: 1.7 μm) column (Waters Corp., Milford, MA, USA), heated at 35 °C and eluted as previously reported with some modifications [61,62]. Briefly, the mobile phase consisted of a solution of acetonitrile (eluent A) and formic acid 0.1% (*v*/*v*) (eluent B). For the elution of sample, a flow rate of 0.3 mL/min was used, and the linear gradient conditions applied were 5–100% A (0–4 min), 100–90% A (4.0–4.1 min), 90% A (4.1–5 min), 90–5% A (5–5.01 min), and 5% A (5.1–8.0 min). The autosampler temperature was set at 4 °C and the injection volume was 10 μL. A Xevo Triple Quadrupole (QqQ) mass spectrometer-based detector (Waters Corp.) was utilised in the MS/MS studies, operating with either positive or negative ionisation electrospray (ESI) (both MS full scan and selected ion recording (SIR) mode were acquired) (Appendix A). The selected multiple reactions monitoring (MRM) mode was used to perform the quantitative study. Prior to the sample analysis, each standard underwent MS manual tuning to optimize the MRM conditions at a concentration of 1 ppm (Appendix A, Appendix A). The following optimum tuning parameters were used to obtain the highest signal levels: 3.0 kV; cone voltage: 36 V; source temperature: 150 °C; dissolution temperature: 500 °C; source dissolution gas flow: 1000 L/h; and gas flow: 20 L/h. High-purity nitrogen gas was utilized as the drying and nebulizing gas, while ultra-high-purity argon was employed as a collision gas. MassLynx software was employed for data collection and processing (version 4.1, Waters Co., Milford, MA, USA).

### 4.5. Total Phenolic Content (TPC) and Total Flavonoid Content (TFC)

The TPC, of each extract, was analyzed through a commercially available polyphenolic quantification assay kit (Folin–Ciocalteu assay) (KB03006, Bioquochem, Asturias, Spain) and performed according to the manufacturer’s instructions. The TPC was determined based on the gallic acid calibration curve (linear range: 0–500 μg/mL, y = 0.005362x + 0.01892, R^2^ > 0.99). The results were expressed as μg of gallic acid equivalents/g of dry extract. The quantification of TFC was performed as it was previously reported with some modifications [62,63]. Briefly, 40 μL of the extract was diluted with 120 μL of methanol and mixed with 20 μL of 10% aqueous solution of aluminum trichloride and 20 μL of 0.5 M aqueous solution sodium acetate. The resulting solutions were allowed to stand in darkness at RT for 40 min, and then the absorbance was monitored on a microplate reader (BioTek Instruments, Inc., Winooski, VT, USA) at 415 nm. The TFC was determined based on the rutin calibration curve (linear range: 0–500 μg/mL, y = 0.0001839x + 0.05676, R^2^ > 0.99). The results were expressed as μg of rutin equivalents/g of dry extract.

### 4.6. Total Condensed Tannins Content (TCTC)

The determination of TCTC was performed according to a previously published experimental protocol [62]. Briefly, 500 μL of each reconstituted extract (in 100% methanol), was diluted with 500 μL of 70% acetone. Then, 3 mL of the n-butanol/hydrochloric acid (37%) mixture (95:5% *v*/*v*) was added, and the resulting solution mixtures were heated at 95 °C for 60 min. Upon completion of the reaction, mixtures were allowed to cool at RT, mixed with ammonium iron (III) and sulfate (NH_4_Fe(SO_4_)_2_) 2% (*w*/*v*), and heated for a further 2 h at 70 °C. Eventually, the absorbance of the cooled solution mixture was monitored on a microplate reader (BioTek Instruments, Inc., Winooski, VT, USA) at 550 nm. The TCTC was determined based on a catechin calibration curve (linear range: 10–100 μg/mL, y = 001912x − 0.02036, R^2^ > 0.993). The results were expressed as μg of catechin equivalents/g of dry extract.

### 4.7. Total Monoterpenoid Content (TMC)

The determination of TMC was performed by adopting a previously reported methodology [62,64]. Namely, 200 μL of each reconstituted extract (in 100% methanol) was mixed thoroughly with 1.5 mL chloroform and allowed to stand for 5 min at RT. Then, 100 μL of concentrated sulfuric acid were added and the suspensions were allowed to stand in darkness for 2 h or until precipitation. The mixture was then centrifuged at 2000 rpm for 6 min. The supernatant was decanted, the formed precipitant was taken up in 95% (*v*/*v*) methanol, and the absorbance was monitored on a microplate reader (BioTek Instruments, Inc., Winooski, VT, USA) at 538 nm. The TMC was determined based on a linalool calibration curve (linear range: 0–60 μΜ, y = 0.005074x + 0.003620, R^2^ > 0.995). The results were expressed as μg of linalool equivalents/g of dry extract.

### 4.8. Cell Lines and Culture

Mouse colon adenocarcinoma cell line CT26 was used for the establishment of the tumor model. Cells were cultivated at 37 °C, with 5% CO_2_ and humidity, in a DMEM culture medium containing 10% fetal bovine serum, 100 U/mL penicillin, 100 μg/mL, 2 mM glutamine, and maintained at 30 to 80% confluent by frequent passage under sterile conditions.

### 4.9. Animals and Tumor Models

Experimental protocols involving mice received approval from the Animal Care and Use Committee and all animal experiments were conducted in light of the 3 R’s (replacement, refinement, and reduction). All mice used for the experiments were not subjected to pain or discomfort.

Female BALB/c mice of 6 to 8 weeks of age and approximately 20 to 25 g body weight were used in the experiments. Animals were bred in the facility, housed in polycarbonated cages under a 12 h light/dark cycle and offered food and tap water ad libitum. Two different preclinical protocols, one short-term and one long-term, were designed. In both cases, mice were randomly separated into two groups. Researchers were not blinded to the groups. In the short-term protocol, test mice (n = 9) were administered 100 μL of a mixture of oregano essential oil in corn oil (0.348 g/kg of body weight) orally for 13 days. Only corn oil was given to control animals (n = 9). On the 10th day, 5 × 10^6^ CT26 cells were inoculated subcutaneously in the back of the neck. Mice were euthanized by cervical dislocation on day 17 and developing tumors, blood, and lymph nodes were collected. In the long-term protocol, test animals (n = 9) were administered 100 μL of the tomato sauce–oregano emulsion mixture for 41 days. A mixture of tomato sauce with the emulsion carrier not carrying the essential oil was given to the control group (n = 9). On day 38, CT26 cells were injected as above. The mice were killed on day 45 and tumors were excised. Animals were monitored daily for signs of discomfort and body weight was measured every 3 or 5 days. Mice exhibiting excessive weight loss (>20%) or clear signs of pain and discomfort were to be sacrificed immediately and excluded; however, no animals reached these criteria. In both protocols, tumor incidence was recorded and tumor volume was determined via the modified ellipsoid formula (width^2^ × length)/2.

### 4.10. Western Blot

Protein levels in the tumor were evaluated with Western blot. Excised tumors were homogenized in RIPA buffer (25 mM Tris-Base, 150 mM NaCl, 0.1% *w*/*v* SDS, 0.5% *w*/*v* sodium deoxycolate, 1% *v*/*v* NP40, and 1 mM DTT) supplemented with protease (PMSF 100 μg/mL, Leupeptin 0.5 μg/mL, Aprotonin 0.5 μg/mL, and Pepstatin 1 μg/mL) and phosphatase (1 mM β-glycerophosphate and 1 mM Na_3_VO_4_) inhibitors. Equal amounts of extracted proteins from each tumor were separated into polyacrylamide gels and transferred onto PVDF membrane. Non-specific binding was blocked with 5% non-fat dry milk. Membranes were incubated with primary antibodies (cleaved caspase 3, 9664, 1:1000; PARP1, 9532, 1:1000; Cox-2, 12282, 1:1000; Cell Signaling, Danvers, MA, USA) or b-tubulin (Sigma Aldrich, St. Louis, MI, USA) at 4 °C overnight and with HRP-conjugated anti-rabbit (7074, 1:2000, Cell Signaling) or anti-mouse (7076, 1:2000, Cell Signaling) antibodies at room temperature for one hour. Bands were visualized with ECL chemiluminescent substrate using a ChemiDoc MP Imaging System (Bio-Rad, Hercules, CA, USA).

### 4.11. Enzyme-Linked Immunosorbent Assay (ELISA)

Cytokine production in the tumor was assessed with ELISA. Collected tumors were homogenized with PBS containing protease inhibitors (10 μg/mL Aprotinin, 10 μg/mL Leupeptin, and 10 μg/mL Pepstatin). Triton X-100 was added to the homogenates to a concentration of 1% and they were rapidly frozen at −80 °C, thawed, and centrifuged at 4 °C, 10,000× *g* for 5 min to discard debris. Commercially available ELISA (Invitrogen, Waltham, MA, USA) kits were used to measure the concentration of IFN-γ (88-8314-22), IL12p70 (88-7121-22), TNF-α (88-7324-22), and granzyme B (88-8022-88), according to manufacturer’s guidelines.

### 4.12. Real-Time qPCR

Gene expression alterations in the tumor were investigated using real-time qPCR. Tumors were homogenized directly in NucleoZol (Macherey-Nagel, Düren, Germany) and RNA was isolated according to supplier’s instructions. RNA concentration and integrity were evaluated with a Nanodrop (Thermo Fisher Scientific, Waltham, CA, USA) and agarose electrophoresis, respectively, and cDNA was synthesized with the High-Capacity cDNA Reverse Transcription Kit (Applied Biosystems, Waltham, CA, USA). A StepOne^TM^ Real-Time PCR System (Applied Biosystems) and the KAPA SYBR^®^ FAST qPCR Master Mix (2X) Kit (Kapa Biosystems, Wilmington, MA, USA) were used for the real-time PCR. *GAPDH* and *beta-actin* were the reference genes and relative gene expression was calculated with the 2^−ddCt^ method. Primers are shown in Appendix A.

### 4.13. Statistical Analysis

Group size for the mice experiments was calculated with power analysis using G*Power (https://www.psychologie.hhu.de/arbeitsgruppen/allgemeine-psychologie-und-arbeitspsychologie/gpower, accessed on 10 April 2025) (Heinrich-Heine Universität Düsseldorf), based on the available literature. Statistical analysis was performed with SigmaPlot (v11, Systat Software, San Jose, CA, USA). Tumor volume, cytokine concentration, and gene expression were assessed with Student’s *t*-test. Immunohistochemistry and relative protein expression were analyzed with Wilcoxon’s rank sum test. Results were considered significant when *p* < 0.05, (* *p* < 0.05, ** *p* < 0.01).

## Figures and Tables

**Figure 1 ijms-26-04737-f001:**
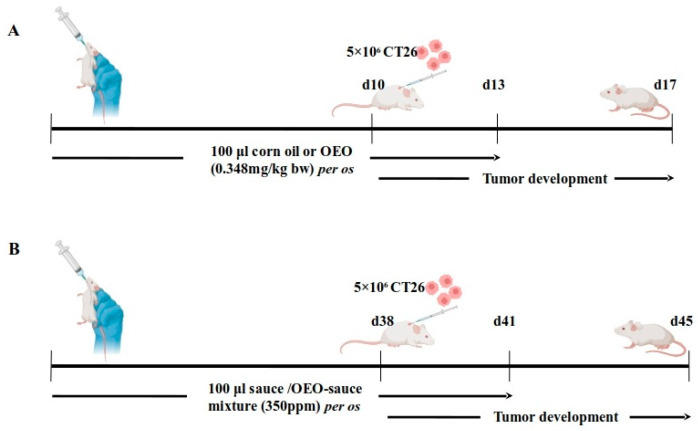
Schematic representation of the preclinical models used. (**A**) Short-term protocol with daily administration of 0.348 mg/kg body weight of essential oil dispersed in 100 μL corn oil. Control mice received corn oil only. (**B**) Long-term protocol with daily administration of 350 parts per million (ppm) essential oil emulsion dispersed in tomato juice. Control group received tomato juice mixed with the emulsion carrier, but without essential oil.

**Figure 2 ijms-26-04737-f002:**
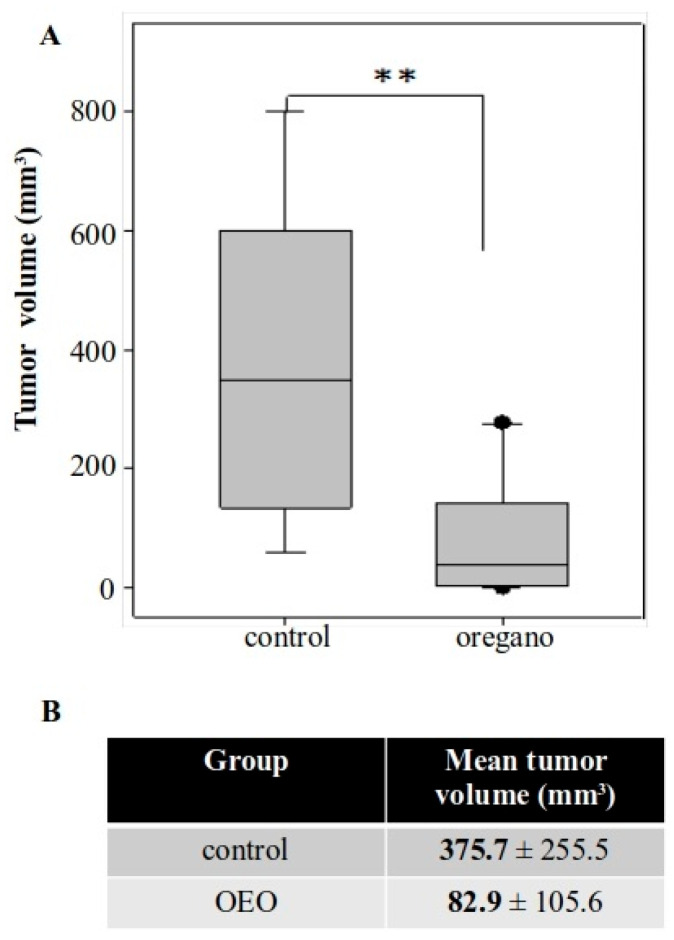
Short-term administration of *Origanum vulgare* ssp. *hirtum* essential oil suppressed subcutaneous colorectal tumor development in BALB/c mice. (**A**) Boxplot of tumor volume from both groups. (**B**) Mean tumor volume was approximately 78% smaller in mice consuming the essential oil. ** *p* < 0.01.

**Figure 3 ijms-26-04737-f003:**
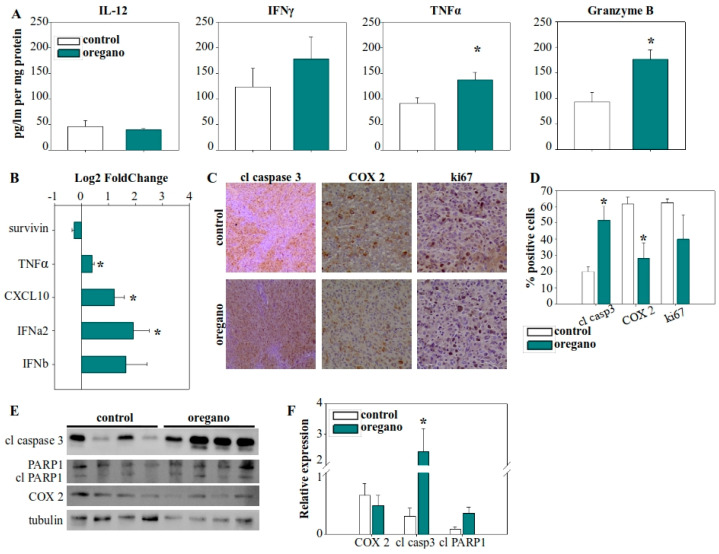
Consumption of the essential oil modulated the immune landscape of the tumor microenvironment, leading to enhanced tumor cell killing. (**A**) Cytokine concentrations in tumor lysates from corn oil- and essential oil-treated mice. (**B**) Gene expression of cytokines and apoptosis-regulating genes in the tumor tissue. GAPDH was used as the housekeeping gene and relative expression levels were calculated using the 2^−ddCt^ method, with control tumors serving as the reference. (**C**) Immunohistochemical analysis of caspase 3 activation, and cyclooxygenase 2 (COX-2) and Ki67 expression. (**D**) Statistical analysis of the percentage of positive cells for cleaved caspase 3, COX-2, and Ki67 from immunohistochemistry slides. (**E**) Investigation of caspase 3 and PARP1 cleavage, as well as COX-2 protein expression in tumor lysates using Western blot. (**F**) Densitometric analysis of cleaved caspase 3, cleaved PARP1, and COX-2 relative expression from Western blot images. Relative expression was normalized to tubulin expression for each sample. * *p* < 0.05.

**Figure 4 ijms-26-04737-f004:**
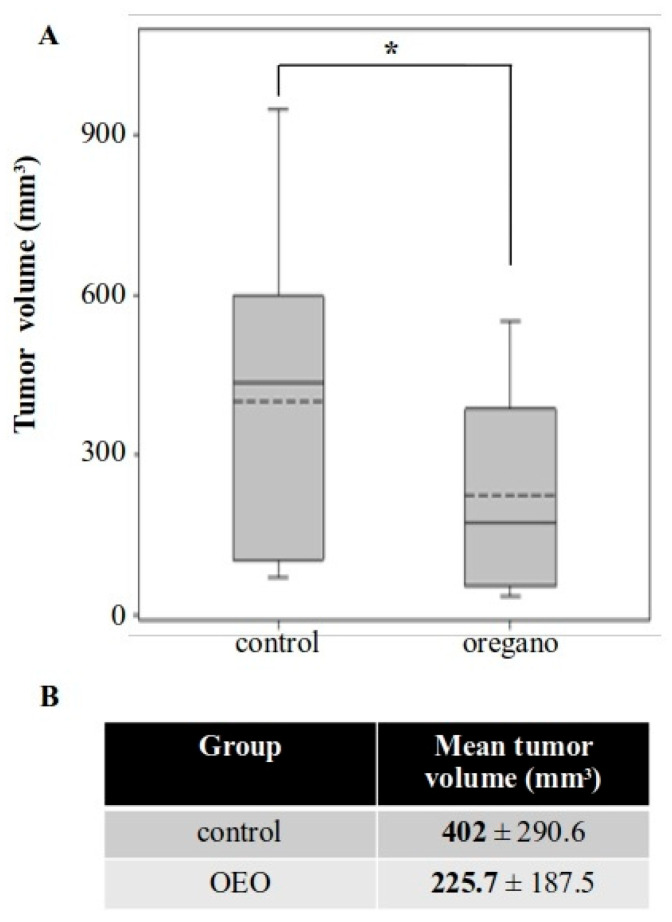
Long-term consumption of tomato juice containing a low concentration of oregano essential oil emulsion inhibited the growth of subcutaneous CT26 tumors. (**A**) Boxplot of tumor volume in control and oregano emulsion groups. (**B**) Mean tumor volume was reduced by approximately 44% in mice administered with the essential oil emulsion. * *p* < 0.05.

**Figure 5 ijms-26-04737-f005:**
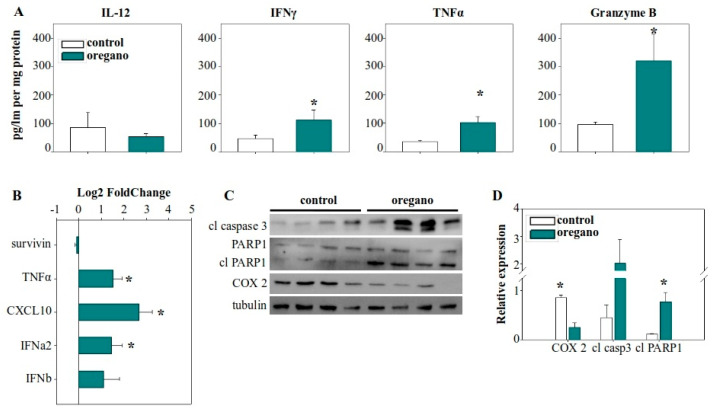
Administration of the tomato sauce supplemented with a low dose of essential oil emulsion induced antitumor immune effects, promoting tumor cell elimination. (**A**) Cytokine concentration in the tumor lysate of control and oregano-treated mice. (**B**) Relative gene expression of cytokine genes and *survivin* in the tumor. *GAPDH* was used as the housekeeping gene and relative expression was calculated with the 2^−ddCt^ method, with control tumors serving as the reference. (**C**) Analysis of caspase 3 activation, PARP1 cleavage, and COX-2 protein expression in tumor homogenates with Western blot. (**D**) Densitometric analysis of cleaved caspase 3, cleaved PARP1, and COX-2 relative protein expression from Western blot images. Relative expression was normalized to tubulin expression for each sample. * *p* < 0.05.

**Table 1 ijms-26-04737-t001:** Quantitative data displaying the total phenolic (TPC), flavonoid (TFC), condensed tannins (TCTC), and monoterpenoid (TMC) content found in *Origanum vulgare* ssp. *hirtum*. The data represent means ± standard deviation (SD) of three independent studies.

**Total Phenolic Content (TPC)** (μg of gallic acid eq/g of dry extract)	25,600.65 ± 148.25
**Total Flavonoid Content (TFC)** (μg of catechin eq/g of dry extract)	17,899.24 ± 235.58
**Total Condensed Tannins Content (TCTC)** (μg of catechin equivalents/g of dry extract)	568.10 ± 15.88
**Total Monoterpenoid Content (TMC)** (μg of linalool equivalents/g of dry extract)	110.47 ± 9.43

**Table 2 ijms-26-04737-t002:** Quantitative data displaying the phytochemical composition of *Origanum vulgare* ssp. *hirtum*. Data collections were obtained via UPLC-ESI(±)-QqQ (UPLC-(electrospray ionization-triple quadrupole)) and standardized to two decimal places. The data represent means ± standard deviation (SD) of five independent studies.

*Origanum vulgare* ssp. *hirtum*	
**Compound**	**Quantity (ng/g of Dry Extract)**
**Benzoic acid derivatives**	
*m*-hydroxy benzoic acid	63.16 ± 2.47
Protocatechuic acid	22.04 ± 0.13
Vanillin	2.65 ± 0.16
*p*-hydroxy benzaldehyde	9.49 ± 0.77
**Gallic acid derivatives**	
Gallic acid	85.71 ± 2.32
Ethyl gallate	127.04 ± 6.14
**Cinnamic acid derivatives**	
Ferulic acid	38.98 ± 2.14
Caffeic acid	68.37 ± 4.96
Dihydro caffeic acid	123.47 ± 6.88
Chlorogenic acid	262.55 ± 10.55
**Coumarin derivatives**	
Coumarin	32.76 ± 2.86
*m*-hydroxycoumarin	87.94 ± 6.54
**Phenolic derivative**	
Eugenol	1060.20 ± 23.21
**Furanocoumarin derivative**	
Xanthotoxol	0.14 ± 0.01
**Flavanone derivatives**	
4′-methoxyflavanone	49.79 ± 2.21
Naringin	19.39 ± 1.21
**Flavonol derivatives**	
Isorhamnetin	25.51 ± 1.36
Quercetin-3-*O*-rhamnoside	6.12 ± 0.08
Myricetin-3-*O*-galactoside	22.55 ± 1.36
Myricetin-3-*O*-rhamnoside	9.08 ± 0.42
Kaempferol	3429.59 ± 89.5
**Procyanidin**	
Procyanidin-B2	11.33 ± 0.06

## Data Availability

Data available on request from the authors.

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
