# Peer review of "Evaluating the Chemical Composition and Antitumor Activity of Origanum vulgare ssp. hirtum Essential Oil in a Preclinical Colon Cancer Model"

_ijms, 2025, doi:10.3390/ijms26104737_

Round 1
Reviewer 1 Report
Comments and Suggestions for Authors
Dear authors,
This manuscript is scientifically sound and has the potential to influence future research. However, major corrections are required to ensure its clarity and accuracy
Here are my comments:
Title:
The title does not accurately reflect the content of the manuscript. I strongly recommend changing the title.
-When you focus on measurable effects or actions of a substance, it is preferable to use “activities” rather than “Antitumor Properties”.
-The chemical composition of the EO is forgotten in the title.
I suggest being precise in the title: It would be better to add the words that reflet exactly the experiment done.
Abstract:
Line 18: What do you mean by “Treatment with low doses of the essential oil”? I suggest to authors to numerical clarification to the term “low doses” and specify the EO dosage.
The authors used the following expressions:
Line 8-19: significantly suppressed tumor growth
Line 19: This effect correlated with enhanced expression
Line 21: was notably increased
I recommend to the authors to add numerical clarification
The authors said that "These findings suggest that O. vulgare spp. hirtum essential oil exhibits promising antitumor properties through immune modulation and apoptosis induction " and "apoptosis induction" but they should clarify whether apoptosis is caspase-mediated or involves additional intrinsic/extrinsic pathways
Make any necessary corrections to the summary after correcting the full text
Introduction
Line 37–38: While the in vitro anticancer effects of Origanum spp. are cited, the Authors should give more details on their in vivo novelty.
The introduction is very short. I strongly recommend to the authors develop further this section. Also, I suggest integrating more information of how the dominant identified compounds relate to prior antitumor findings. It would be better to give more details in how these compounds significantly suppressed tumor growth.
Results
Main weaknesses in this section no data related to the cytotoxicity of the EO on in vivo and in vitro. Kindly add information related to this issue.
Kindly add all the chromatograms generated from this work as supplementary data.
Line 109-111: Mice were administered the essential oil daily vial oral gavage prior to CT26 cell inoculation and during the initial days of tumor development, while control mice received only corn oil: The effect of the EO is treatment/prevention model?
For the chemical composition results show phenolic content in µg/g — but there is no discussion linking specific compounds (e.g., eugenol, kaempferol) to known antitumor effects.
For example, Eugenol is reported as major constituent; authors should mention its known cytotoxic/apoptotic role as a standalone agent. Same for others identified compounds.
Discussion:
The authors should improve the discussion and compare with other works:
- The effect of the EO is treatment/prevention model?
- Give more details regarding the identified compounds and their effect on tumor inhibition
- The EO exhibits promising antitumor properties through immune modulation and apoptosis induction " and "apoptosis induction" : The apoptosis is caspase-mediated or involves additional intrinsic/extrinsic pathways? Kindly develop further this idea
Materials and methods
Please give more information regarding the sample sizer justification.
Why did the authors used female mice?
Limited analysis of toxicity/side effects beyond a claim that “no discomfort” (Line 400) was observed. I strongly suggest to the authors to add toxicity data related (cells and mice) to the used EO.
It would be better to give data related to a third group of mice in which we administrate only the ES without developing a tumor.
Author Response
We sincerely thank the Reviewer for their valuable comments and suggestions. All points have been carefully considered, and detailed, point-by-point responses have been provided.
Comment 1:
The title does not accurately reflect the content of the manuscript. I strongly recommend changing the title.
-When you focus on measurable effects or actions of a substance, it is preferable to use “activities” rather than “Antitumor Properties”.
-The chemical composition of the EO is forgotten in the title.
I suggest being precise in the title: It would be better to add the words that reflet exactly the experiment done.
Response 1:
We thank the reviewer for this valuable suggestion. In response, the title has been revised to more accurately reflect the scope of the study, including both the chemical composition of the essential oil and the experimental evaluation of its biological activities. The phrase “antitumor properties” has been replaced with “antitumor activities” to emphasize the measurable outcomes. The new title also specifies the experimental context and focus on immune-mediated effects.
Comment 2:
Line 18: What do you mean by “Treatment with low doses of the essential oil”? I suggest to authors to numerical clarification to the term “low doses” and specify the EO dosage.
Response 2:
We appreciate the reviewer’s suggestion. Numerical clarification has been added to the abstract. Specifically, the dose of the essential oil administered is now explicitly stated to ensure clarity for the reader.
Comment 3:
The authors used the following expressions:
Line 8-19: significantly suppressed tumor growth
Line 19: This effect correlated with enhanced expression
Line 21: was notably increased
I recommend to the authors to add numerical clarification
Response 3:
Thank you for highlighting this. We revised the abstract to include specific quantitative data where appropriate. Numerical values reflecting the extent of tumor suppression, changes in gene expression, and immune activation markers have been incorporated to support the described effects.
Comment 4: The authors said that "These findings suggest that O. vulgare spp. Hirtum essential oil exhibits promising antitumor properties through immune modulation and apoptosis induction " and "apoptosis induction" but they should clarify whether apoptosis is caspase-mediated or involves additional intrinsic/extrinsic pathways
Response 4:
We thank the reviewer for this insightful point. A more detailed explanation of the apoptosis mechanism has been included in the Discussion, as requested in a similar comment. In summary, based on our current data, we emphasize immune-mediated induction of apoptosis, rather than detailing caspase activation or specific intrinsic/extrinsic pathways, which were not directly assessed in this study.
Comment 5: Line 37–38: While the in vitro anticancer effects of Origanum spp. are cited, the Authors should give more details on their in vivo novelty.
Response 5:
We thank the reviewer for this important point and we agree with the reviewer’s observation. The Introduction section has been expanded to better reflect the novelty of in vivo research on Origanum spp., particularly highlighting the limited number of studies exploring the antitumor effects of Origanum vulgare ssp. hirtum essential oil in live animal models.
Comment 6: The introduction is very short. I strongly recommend to the authors develop further this section. Also, I suggest integrating more information of how the dominant identified compounds relate to prior antitumor findings. It would be better to give more details in how these compounds significantly suppressed tumor growth.
Response 6:
Thank you for this recommendation. The Introduction has been substantially expanded to include a more detailed overview of the known antitumor properties of key constituents identified in our essential oil (e.g., eugenol, thymol, kaempferol). This includes their documented mechanisms of action such as apoptosis induction, antioxidant activity, and immunomodulation, supported by relevant literature. This expanded context highlights both the rationale for our study and its contribution to the field.
Comment 7: Main weaknesses in this section no data related to the cytotoxicity of the EO on in vivo and in vitro. Kindly add information related to this issue.
Response 7:
We appreciate the reviewer’s observation. Data from in vivo toxicity testing of essential oil administration in mice have now been included as supplementary figures, and the corresponding results are presented in the revised Results section. Additionally, in vitro cytotoxicity testing of the essential oil was conducted. However, since comprehensive investigation into the absorption, bioavailability, and systemic circulation of EO constituents remains pending, it is challenging to directly associate these in vitro effects with tumor elimination in vivo. Therefore, we have emphasized the immune-mediated mechanisms in our interpretation of the in vivo findings.
Comment 8: Kindly add all the chromatograms generated from this work as supplementary data.
Response 8:
All chromatograms generated during the study have been added as supplementary figures, as requested.
Comment 9: Line 109-111: Mice were administered the essential oil daily vial oral gavage prior to CT26 cell inoculation and during the initial days of tumor development, while control mice received only corn oil: The effect of the EO is treatment/prevention model?
Response 9:
We thank the reviewer for pointing out the ambiguity. The prophylactic use of the essential oil has now been clarified more explicitly in the revised Results section. We apologize for any confusion caused by the original phrasing.
Comment 10: For the chemical composition results show phenolic content in µg/g — but there is no discussion linking specific compounds (e.g., eugenol, kaempferol) to known antitumor effects. For example, Eugenol is reported as major constituent; authors should mention its known cytotoxic/apoptotic role as a standalone agent. Same for others identified compounds.
Response 10:
We thank the reviewer for this valuable suggestion. We have now expanded the Discussion section to include the known cytotoxic and pro-apoptotic activities of the major EO constituents identified in our study, such as eugenol and kaempferol, and their relevance to antitumor effects. We apologize for the initial omission of this important context.
Comment 11: The authors should improve the discussion and compare with other works:
- The effect of the EO is treatment/prevention model?
- Give more details regarding the identified compounds and their effect on tumor inhibition
- The EO exhibits promising antitumor properties through immune modulation and apoptosis induction " and "apoptosis induction" : The apoptosis is caspase-mediated or involves additional intrinsic/extrinsic pathways? Kindly develop further this idea
Response 11:
We thank the reviewer for these thoughtful and constructive comments. In response:
- The prophylactic nature of the experimental model (i.e., administration of essential oil prior to tumor cell inoculation) has been more clearly emphasized in the revised Discussion section. This preventive approach is relevant in evaluating the immune-modulatory potential of the essential oil prior to tumor establishment.
- We have expanded the Discussion to include a comparative analysis with previous studies involving other Origanum species. Additionally, we discuss in greater detail the tumor-inhibitory effects of the major constituents identified in our essential oil sample, including eugenol, kaempferol, and carvacrol, and their previously reported mechanisms such as pro-apoptotic activity, cell cycle arrest, and immunomodulation.
- Regarding apoptosis mechanisms, we have now elaborated on the pathway analysis. Specifically, we observed activation of caspase-3 and cleavage of PARP1 into characteristic fragments, which are hallmarks of the final execution phase of apoptosis. These findings indicate convergence of both the intrinsic and extrinsic apoptotic pathways. While delineating the precise upstream initiation pathway (intrinsic vs. extrinsic) would be of interest, such analysis is complicated by the interconnected nature of these cascades — for example, via Bid protein, which links the extrinsic pathway to mitochondrial-mediated apoptosis.
Furthermore, granzyme B — secreted by cytotoxic T cells and NK cells — can directly cleave caspase-3 or induce mitochondrial membrane disruption, contributing to the execution phase. Given the complexity of the tumor microenvironment, where various cell death pathways including necrosis may coexist due to immune activation and local stress factors, we believe that downstream markers like caspase-3 and PARP1 cleavage provide the most reliable indication of functional apoptosis in this context.
Therefore, although we do not dissect the contribution of each initiation pathway, our data strongly support that the essential oil induces tumor cell death through immune-mediated and caspase-dependent apoptotic mechanisms.
Comment 12: Please give more information regarding the sample sizer justification.
Response 12:
Thank you for this comment. The sample sizes for the animal experiments were determined using power analysis performed with G*Power software. Parameters for the analysis were based on data from similar previously published studies. This information has now been added to the Statistical Analysis section of the manuscript.
Comment 13.: Why did the authors used female mice?
Response 13:
We appreciate the reviewer’s question. The decision to use female mice was based on standard practice in similar preclinical studies and aimed at reducing variability. Using only one sex helps eliminate sex-based physiological differences as a confounding variable, thereby allowing for more consistent data with smaller group sizes, in line with ethical principles of animal use. Given the exploratory nature of this study and the limited number of published in vivo investigations involving Origanum species, we believe this approach was justified to increase the sensitivity of the model to detect potential biological effects of the essential oil.
Comment 14: Limited analysis of toxicity/side effects beyond a claim that “no discomfort” (Line 400) was observed. I strongly suggest to the authors to add toxicity data related (cells and mice) to the used EO.
Response 14:
Thank you for highlighting this point. In response, supplementary figures (Fig. S2 and Fig. S3) have been added, showing:
- Body weight monitoring throughout the experimental period;
- Spleen and liver indices at the endpoint.
No significant differences were observed between essential oil-treated and control mice in any of these parameters, suggesting no overt toxicity from the essential oil administration.
Comment 15: It would be better to give data related to a third group of mice in which we administrate only the ES without developing a tumor.
Response 15:
We fully agree with the reviewer and thank you for pointing this out. A separate toxicity assessment was indeed performed in a group of non-tumor-bearing mice that received the essential oil. In these animals, serum levels of liver enzymes (ALT, AST) were measured to assess systemic toxicity. The results of this evaluation are now included in Supplementary Figure S3, and the corresponding methodology has been described in the revised Supplementary Methods section.
Reviewer 2 Report
Comments and Suggestions for Authors
In this article, a preclinical study using a colon cancer model in BALB/c mice examines the anticancer activity of the essential oil extracted from the plant Origanum vulgare spp. hirtum, commonly known as Greek oregano. After the oil was taken orally as a preventative measure, it was found that the growth of tumors was significantly inhibited, and the tumor microenvironment produced more cytokines, such as IFN-α2, IFN-γ, and TNF-α. It was also followed by heightened activation of caspase 3 and enhancement of granzyme B production, which are indicators of apoptosis induction and anticancer immunological activity. The same effect was also sustained during extended treatment with the oil as an emulsion preparation, albeit at reduced intensity. The results enhance the prospects of oregano as a natural anticancer immunity enhancer.
The article is significant, I believe, as it presents a complete work that couples chemical analysis of a plant product with preclinical assessment of its biological activity, thereby advancing the research on natural products as anticancer drugs. Additionally, it identifies Greek oregano, a widely used and safe substance, increasing the likelihood that it will be used in adjuvant therapy regimens or cancer prevention. By highlighting new opportunities for nutritional therapies, this study bridges the gap between traditional herbal medicine and modern oncology.
If some important issues are resolved, the article might be published.
- Although there has been an upregulation of TNF-α, IFN-γ, and granzyme B, there has been no evidence of a correlation with the presence or activation of specific immune cell types, such as CD8+ T-lymphocytes or NK cells. The robustness of the results can be increased with additional immunohistochemical or flow cytometry studies aimed at cellular identification of the populations being studied.
- The diagrammatic representation of the experimental protocols, particularly with regard to comparison between short-term and long-term administration, appears to be disjointed. A single diagram that encapsulated the experimental steps, administration duration, and observation period of animals would have helped in enhancing the clarity of the experimental design.
- Results from the two experimental protocols (low and high dosage) are presented without quantitative comparison for the relative expression levels of genes or proteins (e.g., caspase-3 and granzyme B). The use of a direct quantitative comparison would enhance conclusions associated with the effectiveness of the low dosage in emulsion.
- It is stated that there were no observable changes in weight and liver and spleen indices, but no relevant tables or figures are given. It would add to the completeness of the report to present such data, particularly if a potential nutritional application of the substance is proposed.
- The use of tomato juice as a dispersion medium for the oregano presents variables that could influence the outcomes (e.g., lycopene). A control group comprising pure emulsion with no essential oil but with tomato juice would further isolate the effect of the essential oil from any synergistic interaction.
Author Response
We sincerely thank the Reviewer for their valuable comments and suggestions. All points have been carefully considered, and detailed, point-by-point responses have been provided.
Comment 1: Although there has been an upregulation of TNF-α, IFN-γ, and granzyme B, there has been no evidence of a correlation with the presence or activation of specific immune cell types, such as CD8+ T-lymphocytes or NK cells. The robustness of the results can be increased with additional immunohistochemical or flow cytometry studies aimed at cellular identification of the populations being studied.
Response 1:
We thank the reviewer for this insightful observation. As noted, although we did not detect significant changes in the numbers of immune cells in the tumor tissue or draining lymph nodes following essential oil administration, we observed increased expression of several key immune-related molecules, including CXCL10, IFN-γ, and granzyme B. These molecules are known to be closely associated with the activation and recruitment of effector immune cells, including CD8⁺ T lymphocytes and NK cells.
Granzyme B, in particular, is a cytotoxic effector molecule primarily produced by these cell types and its accumulation within the tumor microenvironment, alongside elevated levels of IFNs and the observed caspase-3 cleavage, strongly suggests increased cytotoxic immune activity. While these findings point to the activation of immune responses, we agree that identifying the specific immune cell populations involved would greatly strengthen our conclusions.
We have therefore initiated follow-up studies involving immunohistochemical and flow cytometric analyses to characterize immune infiltrates in greater detail. These efforts aim to delineate the phenotypes and activation states of tumor-infiltrating immune cells and clarify their role in essential oil-mediated tumor inhibition.
Comment 2: The diagrammatic representation of the experimental protocols, particularly with regard to comparison between short-term and long-term administration, appears to be disjointed. A single diagram that encapsulated the experimental steps, administration duration, and observation period of animals would have helped in enhancing the clarity of the experimental design.
Response 2:
We appreciate this suggestion and have addressed it by including a new Figure 1, which provides a clear and unified schematic representation of both experimental protocols. This figure outlines the dosing schedule, formulation used (pure vs. emulsified EO), and the duration of administration and observation for each experimental group. The addition of this visual aid is intended to make the experimental design more transparent and accessible to readers.
Comment 3: Results from the two experimental protocols (low and high dosage) are presented without quantitative comparison for the relative expression levels of genes or proteins (e.g., caspase-3 and granzyme B). The use of a direct quantitative comparison would enhance conclusions associated with the effectiveness of the low dosage in emulsion.
Response 3:
Thank you for this important observation. We acknowledge that a direct quantitative comparison between the two experimental protocols was not included. This was a deliberate decision based on the distinct purposes and contexts of the two studies.
The short-term experiment served as an initial proof-of-concept, using a relatively high dose of essential oil administered via gavage to establish antitumor potential. In contrast, the long-term study was designed to assess the sustained effect of a more physiologically relevant, lower dose incorporated into a dietary emulsion. Due to significant differences in formulation, route of administration, and experimental timeline, we considered a direct quantitative comparison to be methodologically inappropriate and potentially misleading.
Our main objective was to assess whether the antitumor effects persisted even when the daily dose was significantly reduced and administered in a dietary context. We have clarified this rationale in the revised Discussion to better reflect the experimental strategy.
Comment 4: It is stated that there were no observable changes in weight and liver and spleen indices, but no relevant tables or figures are given. It would add to the completeness of the report to present such data, particularly if a potential nutritional application of the substance is proposed.
Response 4:
We thank the reviewer for this suggestion. In response, we have added Supplementary Figures S2 and S3, which include:
- Weekly body weight measurements during the experimental period;
- Liver and spleen weight indices at the endpoint;
- Serum levels of liver enzymes (ALT and AST) from a group of non-tumor-bearing mice treated with the essential oil.
These data collectively support our conclusion that the essential oil, at the doses administered, did not induce significant systemic toxicity or organ-specific damage.
Comment 5: The use of tomato juice as a dispersion medium for the oregano presents variables that could influence the outcomes (e.g., lycopene). A control group comprising pure emulsion with no essential oil but with tomato juice would further isolate the effect of the essential oil from any synergistic interaction.
Response 5:
We fully agree and thank the reviewer for this point. In the long-term experiment, the control group did receive tomato juice supplemented with the same emulsion carrier, but without essential oil, in order to control for any potential effects of tomato juice or the vehicle components.
To clarify this for readers, we have revised the Materials and Methods section, the description of the in vivo experiments, and the legend of Figure 1 to clearly specify the composition of the control formulations. We apologize for the initial lack of clarity and have ensured that this important methodological detail is now explicit in the revised manuscript.
Round 2
Reviewer 1 Report
Comments and Suggestions for Authors
Thank you for your answers
Reviewer 2 Report
Comments and Suggestions for Authors
The authors have revised the manuscript. I recommend the publication of the manuscript.